# From past to present: Exploring COVID-19 in Qatar's hemodialysis population across Omicron dominant and pre-Omicron periods

**Abdullah Hamad** *, **Musab Elgaali, Tarek Ghonimi, Mostafa Elshirbeny, Mohamed Ali, Rania Ibrahim, Muftah Othman, Essa Abuhelaiqa, Hany Ezzat, Karima Boubaker, Mohamad Alkadi, Hassan Al-Malki**

Division of Nephrology, Department of Medicine, Hamad Medical Corporation, Doha, Qatar

* ahamad9@hamad.qa

**Data Availability Statement:** All relevant data are within the paper and its Supporting Information files. All the manuscript data publicly available

## Abstract

COVID-19 carries a high risk of morbidity and mortality in dialysis patients. Multiple SARS-CoV-2 variants have been identified since the start of the COVID-19 pandemic. The current study aimed to compare the incidence and outcomes of the COVID-19 Omicron dominant period versus other pre-Omicron period in hemodialysis patients. In this observational, analytical, retrospective, nationwide study, we reviewed adult chronic hemodialysis patients between March 1, 2020, and January 31, 2022. Four hundred twenty-one patients had COVID-19 during the study period. The incidence of COVID-19 due to the Omicron dominant period was significantly higher than other pre-Omicron period (30.3% vs. 18.7%, P<0.001). In contrast, the admission rate to ICU was significantly lower in the Omicron dominant period than in the pre-Omicron period (2.8% vs. 25%, P<0001) but with no significant difference in ICU length of stay. The mortality rate was lower in the Omicron dominant period compared to the pre-Omicron period (2.4% vs. 15.5%, P<0.001). Using multivariate analysis, older age [OR 1.093 (95% CI 1.044–1.145); P<0.0001] and need for mechanical ventilation [OR 70.4 (95% CI 20.39–243.1); P<0.0001] were identified as two independent risk factors for death in hemodialysis patients with COVID-19. In Conclusion, the COVID-19 Omicron variant had a higher incidence and lower morbidity and mortality than pre-Omicron period in our hemodialysis population.

## Introduction

### 1-Background

COVID-19, caused by a single-stranded mRNA virus, emerged in 2019 and was declared a global pandemic by the WHO in March 2020 [1, 2]. Over the past two years, this infectious disease has affected hundreds of millions of people, resulting in at least 5 million deaths [3]. Although it usually affects the respiratory tract, it can involve other organs, such as the kidneys [4], through different mechanisms, including infection of the tubular epithelium and podocytes, cytokine production, cardiac dysfunction, and hypoxemic tubular injury [5–7]. COVID-

without restriction at PLOS ONE data repository: https://datadryad.org/stash/share/ d4Dd4QuqQEd0Yt95JF1fzx_ PEpTJ7uGq9M8bCsesSyU https://hackmd.io/@ AbdullaHamad3/HkvTi0Hvn

**Funding:** The author(s) received no specific funding for this work.

**Competing interests:** All authors have declared that no competing interests exist.

**Abbreviations:** COVID-19, coronavirus disease; ESKD, end stage kidney disease; HD, hemodialysis; CT Value, The cycle threshold value emerges in RT-PCR tests for the coronavirus; ARDS, Acute respiratory distress syndrome; RNA, Ribonucleic acid; SARS CoV-2, severe acute respiratory syndrome coronavirus 2; VOC, variant of concern; RT- PCR, The reverse transcription-polymerase chain reaction test; ICU, intensive care unit; HMC, Hamad Medical Corporation; PCR, Polymerase Chain Reaction; SD, standard deviation; SPSS, Statistical Package for the Social Sciences; PD, Peritoneal Dialysis; Non-MEC, Non-Middle East Countries; nMabs, Neutralizing Monoclonal Antibodies; RRT, renal Replacement Therapy; Nab, Omicron-specific neutralizing antibodies; APC, Article Processing Charge.

19 clinical presentation can vary from no or mild symptoms to pneumonia, respiratory failure from acute respiratory distress syndrome (ARDS), septic shock, or multi-organ dysfunction [8]. The clinical presentation and outcomes of COVID-19 depend on multiple risk factors such as age, ethnicity, and co-morbid conditions (diabetes mellitus, hypertension, cardiovascular disease, chronic kidney disease, etc.) [9]. Patients with chronic kidney disease had a three-fold risk of developing severe COVID-19 infection, and about one-third of end-stage kidney disease (ESKD) patients requiring hospitalization due to COVID-19 pneumonia died [10, 11]. Patients on dialysis are at a heightened risk of COVID-19 infection due to several factors, including reduced immunity resulting from uremia and toxins, the presence of comorbidities, and the necessity of frequent visits to dialysis centers, which often tend to be crowded environments and can contribute to the spread of the infection [12, 13].

Severe acute respiratory syndrome coronavirus 2 (SARS CoV-2) had multiple rapid mutations, due to its large genome RNA and low fidelity RNA polymerase, since the beginning of the COVID-19 pandemic. Between January 2021 to September 2021, four variants of concern (VOC) have been identified (alfa, beta, gamma, and delta) [14–17]. Those subtypes accumulated heavy protein mutations causing more transmissible and less immunogenic variants. In November 2021, World Health Organization announced the fifth new COVID-19 covariant designated as the B.1.1.529, a variant of concern (VOC), and has been named "Omicron" [18]. The new variant was characterized by high infectivity and widespread and was detected in more than 57 countries. Preliminary studies showed relative resistance to the vaccine [19]. The most common symptoms were fever, severe fatigue, a scratchy throat, wet cough, runny nose, diarrhea, headache, and other body aches [20]. The Current SARS CoV-2 real-time reverse-transcription polymerase chain reaction (RT-PCR) diagnostics continue to detect this variant.

In December 2021, Qatar experienced a notable surge in COVID-19 cases attributed solely to the omicron variant. Notably, a prior study conducted in Qatar during the initial wave in 2020 examined the impact of COVID-19 infection on hemodialysis patients, encompassing incidence, severity, and mortality rates. The study revealed a significant increase in incidence, prolonged hospitalizations, and elevated mortality rates among this specific patient group [21]. In this present study, our objective is to compare the incidence and outcomes of COVID-19 between the Omicron dominant period and pre-Omicron period in hemodialysis (HD) patients within Qatar.

## Objectives

### 1-The primary objective

To compare the incidence of COVID-19 infection during the Omicron dominant period to previous pre-Omicron period in chronic ambulatory HD patients in Qatar.

### 2- Secondary objectives

**2a**- To compare mortality, intensive care unit (ICU) admissions, length of stay in ICU, and need for mechanical ventilation support in the Omicron dominant period versus the pre-Omicron period.

**2b**- To compare characteristics of HD patients infected with COVID-19 who died to patients who stayed alive during the study and identify risk factors associated with death.

## Methods

### Study population and design

Dialysis services in the state of Qatar are provided exclusively by Hamad Medical Corporation (HMC). There are seven dialysis units located in 5 cities across the country (Fahad bin Jassim Kidney Center and Hazam Mubairik General Hospital in Doha city, Al Wakrah (2 units), Alkhor, Al Shahaniya, and Al Shamal). This was an observational, analytical, retrospective, nationwide study. We included all adult ESKD patients on HD infected with SARS CoV-2 between March 1, 2020, and January 31, 2022. Patients who had HD for less than three months were excluded from the study. Since the start of the COVID-19 pandemic, all COVID-19 test results have been recorded in a national electronic medical record system, and all the study data were obtained using it. We have taken several steps to ensure the ethical conduct of the study, including obtaining approval from the institutional review board, ensuring the confidentiality of participant information, and minimizing potential risks to participants. Before the Omicron dominant period, the Ministry of Public Health in Qatar only approved RT-PCR assay of nasopharyngeal swab specimens as the gold standard to diagnose COVID-19. Testing was performed at different locations (dialysis centers, emergency departments, and healthcare centers) for acute presentation or as a routine screening for patients with exposure or at high risk [22, 23]. On December 1, 2021, the rapid antigen test became an accepted alternative by the Ministry of Health due to the overwhelming number of cases nationwide. Based on the national genomic surveillance, almost all cases after December 1, 2021, were due to the Omicron variant. Due to the fast rise of Omicron COVID-19 infections, large scale genome sequencing was not possible in the Omicron variant (compared to previous variants), and positive cases from December 2021 onwards were considered Omicron based on representative sampling [24].

Hospitalization was mandatory for HD patients in Qatar in the pre-Omicron period due to the compulsory isolation strategy. However, during the Omicron dominant period, the policy became more lenient, with self-isolation at home acceptable for stable patients. The cycle threshold (CT) value was done routinely on most patients upon diagnosis with COVID-19 in Qatar.

We divided HD patients with COVID-19 into Omicron dominant period and pre-Omicron period. Per national genomic surveillance, all patients diagnosed with COVID-19 before December 1, 2021, were assigned to the pre-Omicron period, while patients diagnosed on December 1, 2021, and onwards were assigned to the Omicron dominant period. Each episode of COVID-19 illness was considered an event and analyzed separately. Patient characteristics, including demographics, comorbidities, vaccination status, clinical features, and outcomes, were collected from the national electronic medical record covering all Qatar health sectors (Cerner-North Kansas City, MO, USA).

### Ethics statement

The study was approved by the Institutional Review Board of Hamad Medical Corporation (MRC-05-161; July 20, 2020). The IRB waived the informed consent given the study's retrospective design. All data were fully anonymized before being accessed and analyzed anonymously.

### Statistical analysis

The incidence of COVID-19 Omicron infection within the dialysis population was determined by dividing the number of Omicron dominant period cases by the total number of dialysis

patients. Similarly, a similar method was employed to calculate the incidence of pre-Omicron COVID-19 infections. We also reported the infection rate, calculated as the number of cases per week per 1000 dialysis patients, to effectively illustrate the spread of COVID-19 over time. Qualitative variables are shown with their frequency distributions. Quantitative variables are summarized as mean ± SD or median and interquartile range. The association between qualitative variables was evaluated using Chi-square or Fisher's exact test. Quantitative variables were analyzed using Student's t-test or analysis of variance. Statistical analyses were performed using SPSS software (version 21.0; Chicago, IL, USA). Statistical significance was defined as a 2-sided P value of <0.05. We used binary logistic regression analysis to analyze risk factors for mortality. We performed univariate analysis first to analyze each variable in our data set, separately. We did it in the beginning to identify single variables that are statistically significant to use them in more advanced tests (multivariate analyses). Afterward, we performed multivariate analysis to compare multiple variables at a time. This offers a more complete examination of data by looking at all possible independent variables and their relation to one another.

In Table 1, we used chi square test to determine p value and potential significant factors, while in Table 3 we used univariate and multivariate regression analysis to determine odds ratio and 95% confidence intervals for those factors.

## Results

### 1- Patients characteristics and clinical presentation

Between March 1, 2020, and January 31, 2022, 421 out of 900 HD patients had COVID-19 (46.8%);26 were infected with both Omicron dominant period and pre-Omicron period. The mean age of patients was 59 ±16.1 years. Most patients were males (n = 287; 64%), and hypertension was the most frequent comorbidity (n = 412; 97.9%), followed by diabetes mellites (n = 282; 67%). Most patients received at least two doses of the COVID-19 vaccine (n = 263; 62.5%), and just about half received the third booster dose (n = 209; 49.6%). BioNTech and

**Table 1. Clinical and demographic characteristics of surviving and non-surviving patients.**

| Factors | Dead (n = 33) | Alive (n = 388) | P value |
|---|---|---|---|
| Age | 71.3±10.8 | 58±16.1 | <0.001 |
| MEC/nonMEC | 9/24 (27/73) | 173/215 (44.5/55.5) | 0.067 |
| Gender (male/female) | 26/7 (78.7/21.2) | 247/141(63.6/36.4) | 0.081 |
| Diabetes | 23 (69.6) | 259 (66.7) | 0.730 |
| Hypertension | 33 (100) | 379 (97.6) | 0.376 |
| Cardiovascular diseases | 18 (54.5) | 165 (42.5) | 0.181 |
| Obstructive lung diseases | 2 (6.0) | 10 (2.5) | 0.248 |
| Hospitalization | 32 (96.9) | 245 (63.1) | <0.001 |
| ICU admission | 25 (75.7) | 26 (6.7) | <0.001 |
| Mechanical ventilation | 22 (66.6) | 9 (2.3) | <0001 |
| No COVID-19 vaccination | 23 (69.6) | 133 (34.2) | <0.001 |
| Vaccination 1st dose | 10 (30.3) | 253 (65.2) | <0.001 |
| Vaccination 2nd dose | 9 (27.2) | 252(64.9) | <0.001 |
| Vaccination booster dose | 5 (15.1) | 204 (52.2) | <0.001 |
| Omicron virus | 6 (18) | 241 (36.3) | <0.001 |
| Pre-Omicron virus | 27 (81.8) | 147 (37.8) | <0.001 |
| CT value | 16.2±10 | 18.5±8.9 | 0.161 |
| ICU length of stay | 15.2±20.4 | 11.3±10.8 | 0.351 |

**Table 2. Baseline demographic and clinical characteristics of COVID-19 hemodialysis patients study population in Qatar.**

| | All (n = 421 (%) | Omicron dominant period (n = 247) | Pre-Omicron period (n = 174) | P value |
|---|---|---|---|---|
| Age (years) | 59 ±16.1 | 59.1±16.5 | 58.9±15.6 | 0.855 |
| Gender:<br>male/female | 274/147 (65.1/34.9) | 155/92 (62.6/37.4) | 119/55 (68.6/31.4) | 0.220 |
| Nationality:<br>MEC/nonMEC | 182/239 (43.3/56.7) | 122/125 (49.4/50.6) | 60/114 (34.4/65.5) | 0.002 |
| Comorbidities:<br>Diabetes:<br>Hypertension:<br>Cardiovascular Disease:<br>Obstructive lung diseases: | 282 (67)<br>412 (97.9)<br>183 (43.5)<br>12 (2.9) | 175 (70.4)<br>241 (97.5)<br>139 (65.2)<br>9 (3.6) | 107 (61.4)<br>170 (97.7)<br>44 (25.2)<br>3 (1.7) | 0.048<br>0.853<br><0.001<br>0.241 |
| Vaccination:<br>1st dose<br>2nd dose<br>Booster dose | 263 (62.4)<br>261 (61.9)<br>209 (49.6) | 211 (85.0)<br>210 (84.6)<br>209 (84.2) | 52(29.8)<br>51 (29.3)<br>0 (0) | <0.001<br><0.001<br><0.001 |
| Vaccine name:<br>Pfizer<br>Moderna<br>Non vaccinated | 223 (52.9)<br>42 (9.9)<br>156 (37.0) | 181 (73.4)<br>32 (12.9)<br>34 (13.7) | 42 (24.1)<br>10 (5.7)<br>122 (70.1) | <0.001 |
| Previous covid infection | 26 (6.1) | 26 (10.1) | 0 (0) | <0.001 |
| CT value | 18.3±9.1 | 20.9±5.0 | 15.9±11.1 | <0.001 |
| Hospitalization | 277 (65.7) | 113 (45.7) | 164 (93.6) | <0.001 |
| ICU admission | 51 (12.1) | 7 (2.8) | 44 (25.2) | <0001 |
| ICU length of stay (Days) | 13.1±15.9 | 15.7±8.6 | 12.8±16.7 | 0.657 |
| Mechanical ventilation | 31 (7.3) | 3 (1.7) | 28 (16.0) | <0.0001 |
| Mortality | 33 (7.8) | 6 (2.4) | 27 (15.2) | <0.001 |

Pfizer's vaccines were the only utilized vaccines. Fig 1 show flow diagram for dialysis patients with COVID-19 in the state of Qatar between March 2020 to January 2022. Table 2 summarizes the demographics and general characteristics of the study population.

## 2- Primary outcome

The infection rate was much higher during the Omicron dominant period [247 cases in 2 months (30.7 cases/week/1000 HD patients)] compared to the whole pre-Omicron **period** waves [174 cases in 21 months (2.33 cases/week/1000 HD patients)] overall, incidence of Omicron dominant period infections in our dialysis population was higher than the pre-Omicron period (30.3% vs. 18.7%, p <0.001) (Fig 2). The pre-Omicron period extended over 21 months with small spikes, the highest being 39 cases in May 2020. In contrast, the Omicron dominant wave was short, surging quickly to 225 cases in January 2022. The male ratio showed no significant difference between the pre-Omicron and the Omicron dominant period waves (62.6% vs. 68.6%, P = 0.220).

**Table 3. Factors associated with increased risk of mortality in COVID-19 hemodialysis population in the state of Qatar.**

| | Univariate analysis | | | Multivariate analysis | | |
|---|---|---|---|---|---|---|
| Risk Factors | Odds ratio | 95% CI | P | Odds ratio | 95%CI | P |
| Age | 1.067 | 1.063–1.099 | 0.0001 | 1.093 | 1.044–1.145 | **0.0001** |
| Ventilation | 84.222 | 31.603–224.452 | <0.0001 | 70.422 | 20.395–243.160 | **0.0001** |
| Hospitalization | 18.286 | 2.472–135.270 | 0.004 | 6.743 | 0.823–55.255 | 0.075 |
| First dose vaccination | 0.232 | 0.107–0.502 | 0.0001 | 1.101 | 0.301–4.035 | 0.884 |

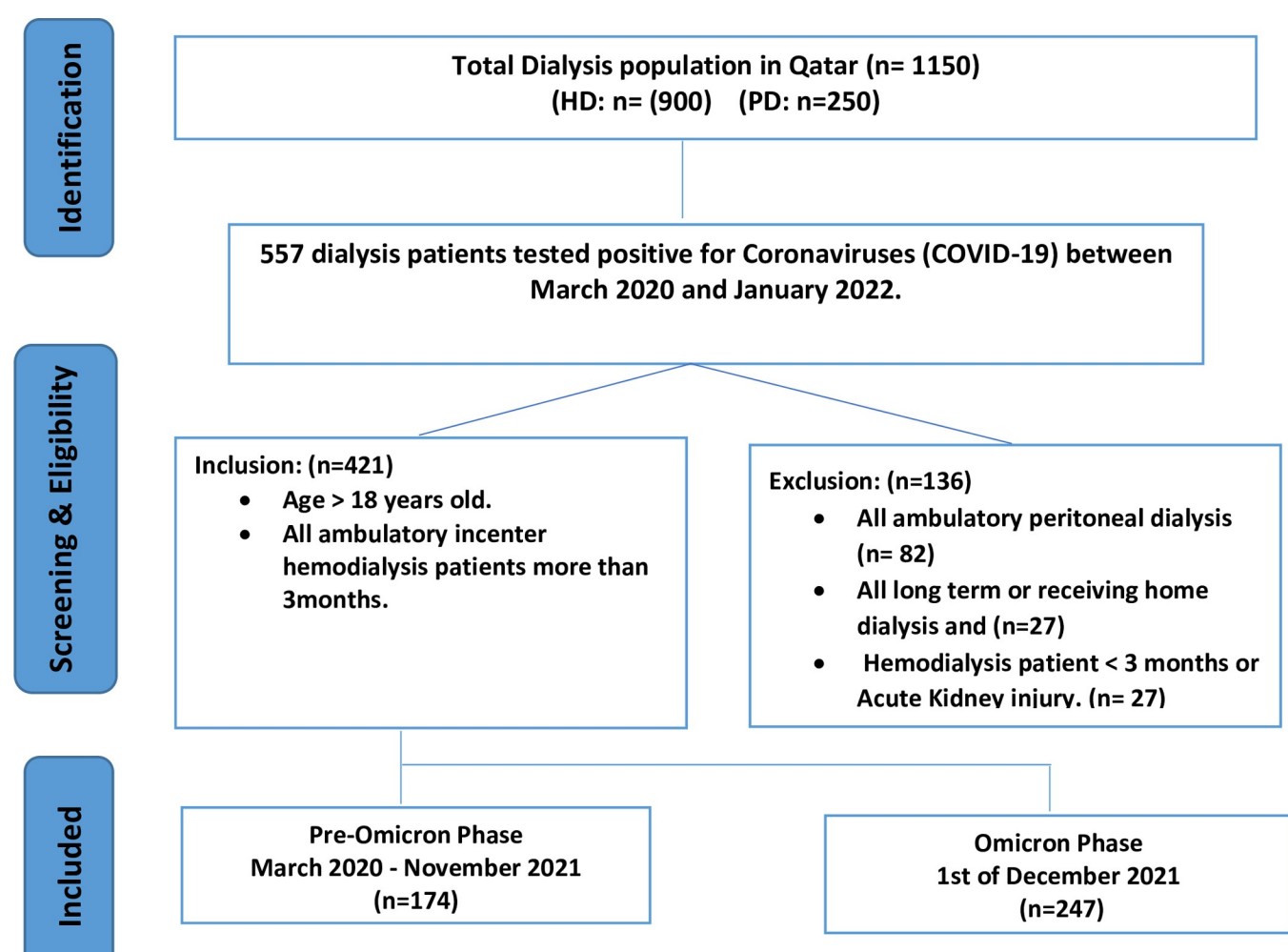

COVID-19, Corona virous infection; HD, patients receiving Hemodialysis; PD, patients receiving Peritoneal dialysis.

**Fig 1. Flow diagram for dialysis patients with COVID-19 in the state of Qatar between March 2020 to January 2022.**

During the pre-Omicron period, non-Middle East Countries (non-MEC) patients represented 66% of reported COVID–19 infections, compared to 50% for the period of Omicron dominant period wave (P = 0.002). The Omicron dominant period has affected more patients with cardiovascular disease (65.2% vs. 25.2%; P<0.001) and Diabetes Mellitus (70.4% vs. 61.4%, P = 0.048) (Table 2).

## 3- Secondary outcomes

**3.a Comparison of morbidity and mortality in Omicron dominant period VS pre-Omicron period.** CT value was significantly higher throughout the Omicron dominant period (20.9±5) compared to the pre-Omicron dominant period (15.9±11.1) (P < 0.001). More

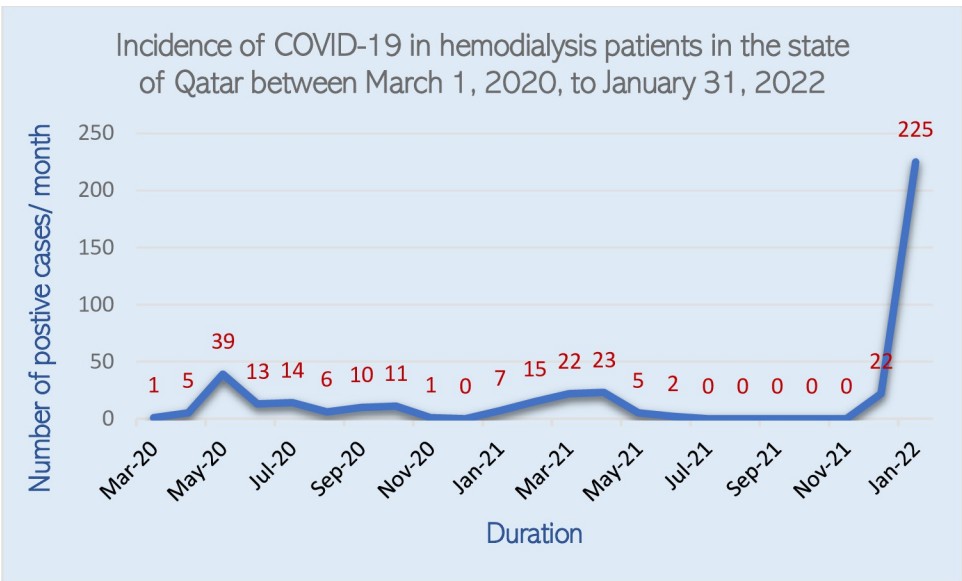

**Fig 2. Monthly new cases (incidence) of COVID-19 in hemodialysis patients a comparison between (non-Omicron, and Omicron variant wave).**

patients required hospitalization in the pre-Omicron period versus the Omicron dominant period (93.6% vs. 45.3% (p<0.001). In the same context, the ICU admission rate was significantly higher with pre-Omicron period vs. Omicron dominant period (25.2% vs. 2.3%. p<0001); nevertheless, no significant difference was recorded in the ICU length of stay 15.7 ±8.6 vs. 12.8±16.7 (p = 0.657). The mortality rate was much higher in the pre-Omicron period versus the Omicron dominant period (15.2% vs. 2.4%. P<0.001).

**3.b Mortality.** Among overall 421 infection episodes, there were 33 deaths due to COVID–19 infections (non-survivors). Deaths mostly occurred during the pre-Omicron period compared to the Omicron dominant period (27 vs. 6; P<0.001), with a significantly higher mean age (71.3 vs 58.16 P<0.001). Although male patients had more deaths (78% of males in the death group vs. 63% in the alive group) but it was not statistically significant (P = 0.08). Death was significantly higher in those who were hospitalized (96.6% vs. 63.1%; P<0.001), admitted to ICU (75.7% vs. 6.7%; P<0.001), received mechanical ventilation (66.6% vs. 2.3%; P <0.001) and did not receive COVID vaccine (69.6% vs. 34.2%; P<0.001).

**3.c Risk factors for death in HD patients with Covid-19.** Some traits were associated with increased mortality among dialysis patients with COVID-19, i.e., age [OR = 1.067 (95% CI 1.063–1.099), P = 0.001], need for mechanical ventilation [OR = 84.2 (95% CI 31.6–224.4), P <0.0001], and hospitalization [OR 18.2 (95% CI2.47–135.2), P 0.004]. Other factors approached but did not reach statistical significance such as male gender [OR = 2.47 (95% CI 0.99–6.15), P = 0.051], CT value [OR = 0.984 (95% CI 0.910–1.050), P = 0.688] and length of stay in ICU [OR = 1.003 (95% CI 0.357–1.050), P = 0.915]. First-dose vaccination was protective in the univariate analysis [OR of 0.23 (95% CI 0.107–0.502), P = 0.0001], but was not on multivariate analysis [OR 1.1 (95% CI 0.301–4.035), P = 0.88].

Multivariate analysis confirmed only two statistically significant risk factors. Age posed a small but significant risk [OR 1.093 (95% CI 1.044–1.145), P<0.0001], but most importantly, was a requirement for mechanical ventilation which tremendously increased the risk of death [OR 70.4 (95% CI 20.39–243.1), P<0.0001]. Table 3 summarizes the factors associated with increased mortality risk in the COVID-19 HD population in Qatar.

## Discussion

In this observational, analytical, retrospective, nationwide study, we followed dialysis patients over almost two years and compared the incidence and complications of COVID-19 Omicron dominant period to pre-Omicron period. Our results revealed a significantly higher incidence of Omicron dominant period infections than in the pre-Omicron era (30.3% vs. 18.7%, p <0.001). However, despite the higher infection rate, the mortality rate was much lower in the Omicron dominant period than pre-Omicron period (2.4% vs. 15.2%; P<0.001). Similarly, other complications aspects of COVID-19 infection in the Omicron dominant period, like the severity of disease and ICU admissions, were significantly lower than in the pre-Omicron period.

Management of COVID-19 cases in Qatar evolved with the changes in the virus. In pre-Omicron period, it followed strict isolation policies, admitting all cases to one designated hospital for COVID-19, including ICU. As the disease burden decreased with the Omicron dominant period wave, the isolation policies became less tightened, with patients spread across all hospitals. The same applied to dialysis; all cases were admitted initially (pre-Omicron) to the hospital then, gradually, patients were distributed to other hospitals or could stay home and go to designated HD facilities [21–31].

Our data on improved outcomes in the setting of hospitalization and mortality in the era of Omicron infections is consistent with previous studies with the same results. A recent population study from Scotland revealed an improvement in mortality rate from 26.7% in pre-omicron infections to 4% in omicron infections but with increased transmission of infection and dramatic rises in SARS-CoV-2 infected patients [32]. Another study from the United Arab Emirates revealed that despite nearly all patients receiving vaccination and booster doses, the infection rate exceeded previous waves in terms of the number of cases and the speed of transmission. Remarkably, the study also found that the Omicron variant showed significantly lower morbidity and mortality, highlighting its reduced danger compared to previous variants [33].

In our study, the CT value was significantly lower in the pre-Omicron period (15.9±11.1) compared to the Omicron dominant period (20.9±5) (P < 0.001). Although the rapid test was permitted for diagnosis, most patients in our Omicron cohort had RT-PCR (65%). The difference in CT value eventually did not translate to be a significant risk factor for mortality in our population.

Omicron is manifested by high transmission for multiple reasons [34]. Cellular and humoral immunity responses are lower against the Omicron variant than pre-Omicron in general populations [35, 36] and in dialysis patients [37]. Of interest, Omicron-specific neutralizing antibodies (Nab) titer was considerably lesser than Delta-specific NAb, which may explain the more effective escape of the Omicron virus from neutralizing antibodies and its efficient spread among even in vaccinated people [38, 39].

Recent studies have shown that the booster dose of SARS-CoV-2 vaccines, especially mRNA-based vaccines, has proven to be relatively effective in providing protection against Omicron infection within the general population [40–42] (despite the vaccine evasion mentioned above). This benefit also extends to patients with kidney disease [43–46]. Notably, almost 50% of our patients received the third vaccine dose, which could partially account for the milder disease course observed compared to previous variants.

Another explanation is the introduction of neutralizing monoclonal antibodies (nMabs) and oral antiviral therapy which started to be widely used in symptomatic patients receiving RRT (Dialysis or transplantation) following Covid-19 infection. A third explanation may be the viral strain that produces considerably attenuated disease. Based on the 36 mutations

noted in the spike protein in the Omicron variant, antibody responses could have been extremely suboptimal compared to previous variants [47].

Our study demonstrated two main risk factors of mortality in dialysis patients who experienced COVID-19 infection: age [OR 1.093 (95% CI 1.044–1.145; p 0.0001)] and need for mechanical ventilation [OR 70.4 (95% CI 20.39–243.1; P 0.0001]. This is similar to previous studies that showed high mortality and worse outcomes among HD patients with COVID-19 [48–50]. Our results of increased mortality in older dialysis patients due to COVID-19 infection are consistent with some previous studies [51, 52]. Our data suggested that the male gender might have a higher mortality risk (although it approached but did not reach statistical significance), consistent with the findings of a large study done at the national level in the United States [53]. According to another study conducted in New York, dialysis patients with COVID-19 were more likely to be older, male, Black, or Hispanic. Additionally, these patients were found to have a longer duration of maintenance dialysis [54]. Old age and male sex also were significant mortality risk factors in dialysis patients infected with covid-19 in another trial [55]. Based on all these findings, we may need to focus more concentrations to this subgroup of patients with a high risk of mortality among dialysis patients.

The strengths of our analysis include the diverse nature of our patients by age, gender, ethnicity, socioeconomic status, and dialysis modality and the availability of detailed information on the characteristics and outcomes of hospitalized patients as we use the same system in all healthcare facilities. It is the first national study in Qatar to compare the outcome of Omicron dominant period vs. pre-Omicron COVID-19 period infection among hemodialysis patients.

This study had some limitations. First and most important, not all cases of Omicron dominant period are tested with genomic sequencing to prove the diagnosis. But sample testing shown that majority of COVID-19 cases in that period were of Omicron variant [24]. Second, our study has a limited number of patients, but it represented a whole country cohort (hemodialysis patients in Qatar). Third, we did not measure antibody titers during our research. However, we depended on the rate of confirmed infection and mortality rate in comparison between Omicron dominant period infection and pre-Omicron period (which should be more accurate). And finally, we could not confirm the actual effect of COVID-19 booster vaccination on reduced morbidity and mortality as it was used almost simultaneously as antiviral medications.

## Conclusion

We present the first national study to compare the outcome of Omicron vs. pre-Omicron COVID-19 variants infection among hemodialysis patients. Although the incidence of Omicron variant infection was much higher than pre-Omicron period, mortality, and ICU admission were significantly higher in the pre-Omicron period compared to the Omicron variant. Furthermore, the need for mechanical ventilation was the strongest predictor of mortality in our whole Covid-19 infected HD population.

## Supporting information

**S1 File. Coded master sheet Omicron dominant.**
(CSV)

**S2 File. Coded master sheet pre-Omicron dominant.** https://datadryad.org/stash/share/ d4Dd4QuqQEd0Yt95JF1fzx_PEpTJ7uGq9M8bCsesSyU. https://hackmd.io/@AbdullaHa- mad3/HkvTi0Hvn.
(CSV)

**S1 Checklist. STROBE statement—checklist of items that should be included in reports of observational studies.**
(DOCX)

## Acknowledgments

We profusely thank all the contributors from HMC for their excellent efforts and continued support; Mathew M, Ateya H, Aly S, Singh P, Yasin F, Zitouni Y, Al-Mohanadi T, Joseph Sh., and all the Hamad Medical Corporation staff for their devotion to patient care during the COVID-19 pandemic.

Part of this manuscript was presented in an abstract form at the American Society of Nephrology meeting in Orlando, FL 2022 [56].

## Author Contributions

**Conceptualization:** Abdullah Hamad, Musab Elgaali, Tarek Ghonimi, Mostafa Elshirbeny, Rania Ibrahim, Mohamad Alkadi.

**Data curation:** Abdullah Hamad, Tarek Ghonimi, Muftah Othman, Essa Abuhelaiqa, Hany Ezzat, Karima Boubaker.

**Formal analysis:** Abdullah Hamad, Tarek Ghonimi.

**Methodology:** Abdullah Hamad, Musab Elgaali, Tarek Ghonimi, Mohamed Ali, Rania Ibrahim, Mohamad Alkadi.

**Project administration:** Abdullah Hamad.

**Resources:** Mostafa Elshirbeny, Mohamed Ali, Rania Ibrahim, Muftah Othman, Essa Abuhelaiqa, Hany Ezzat, Karima Boubaker.

**Supervision:** Hassan Al-Malki.

**Validation:** Abdullah Hamad.

**Writing – original draft:** Abdullah Hamad, Musab Elgaali, Mostafa Elshirbeny, Mohamed Ali, Rania Ibrahim.

**Writing – review & editing:** Mohamad Alkadi, Hassan Al-Malki.

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
