## [Decision Letter · Decision Letter 0]

5 Jun 2023

PONE-D-23-13125A Comparative Analysis of COVID-19 Omicron and Non-Omicron Variants in Hemodialysis Population of QatarPLOS ONE

Dear Dr. Hamad,

Thank you for submitting your manuscript to PLOS ONE. After careful consideration, we feel that it has merit but does not fully meet PLOS ONE’s publication criteria as it currently stands. Therefore, we invite you to submit a revised version of the manuscript that addresses the points raised during the review process.

The manuscript needs major revision regarding the Tables, some results and the discussion part in order to justify the conclusion.

We look forward to receiving your revised manuscript.

Kind regards,

Fadi Aljamaan

Academic Editor

PLOS ONE

Journal Requirements:

https://www.ajkd.org/article/S0272-6386(21)00771-X/fulltext

https://www.tandfonline.com/doi/abs/10.1080/17512433.2021.1902303?journalCode=ierj20

https://pubmed.ncbi.nlm.nih.gov/35246846/

https://jamanetwork.com/journals/jamanetworkopen/fullarticle/2786199

https://www.asn-online.org/education/kidneyweek/2022/program-abstract.aspx?controlId=3767798

In your revision ensure you cite all your sources (including your own works), and quote or rephrase any duplicated text outside the methods section. Further consideration is dependent on these concerns being addressed.

"All authors have declared that no competing interests exist."

6. Please amend either the abstract on the online submission form (via Edit Submission) or the abstract in the manuscript so that they are identical.

7. Your ethics statement should only appear in the Methods section of your manuscript. If your ethics statement is written in any section besides the Methods, please move it to the Methods section and delete it from any other section. Please ensure that your ethics statement is included in your manuscript, as the ethics statement entered into the online submission form will not be published alongside your manuscript. 

Additional Editor Comments:

Dear Authors

I reviewed the manuscript as a second reviewer in order to expedite the review process and send decision letter in a timely manner. Please note the comments of mine and the other reviewer.

Reviewers' comments:

Reviewer's Responses to Questions

**Comments to the Author**

1. Is the manuscript technically sound, and do the data support the conclusions?

Reviewer #1: Yes

Reviewer #2: Partly

2. Has the statistical analysis been performed appropriately and rigorously? 

Reviewer #1: Yes

Reviewer #2: No

3. Have the authors made all data underlying the findings in their manuscript fully available?

Reviewer #1: Yes

Reviewer #2: No

4. Is the manuscript presented in an intelligible fashion and written in standard English?

Reviewer #1: Yes

Reviewer #2: No

5. Review Comments to the Author

Reviewer #1: I would highly recommend if not mentioned the isolation precautions followed in the hemodialysis units as this can be of great value to critical appraise this article

May add more specific info like is inside the hospital hemodialysis or separate centers

Reviewer #2: Thanks for that interesting paper that addressed a recent variant of the SARS CoV2 namely Omicron in comparison to the previous variants.

You need to address some points in order to make the manuscript publishable.

1. Please review Table 1 regarding the numbers as they do not sum up correctly and some percentages are not correct.

2. There is no Table addressing the characteristics on Omicron versus non Omicron please tabulate the data.

3. How did you compare CT value between both groups even though RT-PCR was not used to detect virus since early December 2021.

4. You mentioned clearly in the Mortality section that Omicron related infections caused significantly higher mortality compared to non-Omicron, even though I do not see it in the Univariate and multivariate analysis table.!!

5. there was significant difference in all doses of vaccination between those who died and who did not, how did vaccination behave in the factor analysis.

6. Please describe how COVID was managed in Qatar during the Omicron and on Omicron period, in ICU and general ward.

7.You mention use on monoclonal Ab and anti viral as possible explanation for the lower mortality in Omicron infected patients, was that standard regime of treatment of COVID during that time, how many patients received it?

8. Your argument for the neutralizing Ab mounted from Omicron infection in the discussion is confusing, do you advocate that it contributed to lower mortality or not?

9. As Omicron caused much milder disease and lower mortality, on what basis do you advocate to deliver 4th booster dose.

10. The manuscript needs English language editing please.

6. PLOS authors have the option to publish the peer review history of their article (what does this mean?). If published, this will include your full peer review and any attached files.

Reviewer #1: **Yes: **Mohammed alarifi

Reviewer #2: **Yes: **Aljamaan, Fadi

---

## [Author Response · Author response to Decision Letter 0]

29 Jun 2023

Dear Editor in Chief:

Thank you for your valuable comments. Hopefully, we have addressed all comments properly. We believe these comments led to significant improvement in our manuscript.

Reviewers' comments:

A- Please ensure that your manuscript meets PLOS ONE's style requirements, including those for file naming.

Thank you for your comment. We did the required modifications on the manuscript template according to PLOS ONE's style requirements.

B- Competing Interests section.

Thank you for your comment. We updated the competing Interests section in the revised manuscript, cover letter, online submission form, and editorial manager.

C- Please note that in order to use the direct billing option the corresponding author must be affiliated with the chosen institute. Please either amend your manuscript to change the affiliation or corresponding author or email us at plosone@plos.org with a request to remove this option.

Thank you for your comment. Please note that the corresponding author Dr. Abdullah Hamad is affiliated with Hamad Medical Corporation, and the submission form has been updated accordingly. 

D- We note that you have stated that you will provide repository information for your data at acceptance. Should your manuscript be accepted for publication, we will hold it until you provide the relevant accession numbers or DOIs necessary to access your data. If you wish to make changes to your Data Availability statement, please describe these changes in your cover letter and we will update your Data Availability statement to reflect the information you provide

Thank you for your comment. We uploaded the raw data (Coded Omicron VS pre Omicron Master sheet, supporting information, README.md file, IRB approval) on the PLOS1 open data with the following DOI: https://doi.org/10.5061/dryad.9w0vt4bmn

E- Please amend either the abstract on the online submission form (via Edit Submission) or the abstract in the manuscript so that they are identical.

Thank you for your comment. We amended the abstract in the revised manuscript.

F- Your ethics statement should only appear in the Methods section of your manuscript. If your ethics statement is written in any section besides the Methods, please move it to the Methods section and delete it from any other section. Please ensure that your ethics statement is included in your manuscript, as the ethics statement entered into the online submission form will not be published alongside your manuscript. 

Thank you for your comment. The ethics statement is now included in the methods section only.

G- We noticed you have some minor occurrence of overlapping text with the following previous publication(s), 

Thank you for your comment. We made all the necessary changes as advised. We also added two references (37 and 56), our abstracts presented in the American Society of Nephrology 2022.

1- Please review Table 1 regarding the numbers as they do not sum up correctly and some percentages are not correct.

Thank you for your comment. We reviewed and corrected the numbers in Table 1.

2. There is no Table addressing the characteristics on Omicron versus non-Omicron please tabulate the data.

Thank you for your comment. We have all characteristics in Table 1 (including demographics, co-morbidities, outcomes, etc.). If you feel we should have it separate, we will gladly do so.

3. How did you compare CT value between both groups even though RT-PCR was not used to detect virus since early December 2021. 

Thank you for your comment. Although COVID-19 rapid antigen test was allowed to be used in the Omicron era, most patients had RT-PCR done. There were 160 out of 247 (65%) in the Omicron group, and 125 out of 174 (72%) in the non-Omicron group who had RT-PCR, and their CT values were reported. Although all 174 patients in the non-Omicron group had RT-PCR, CT values were unavailable for 49 patients, as the lab did not report CT values early during the COVID-19 pandemic. We included this information in the discussion section to clarify.

4. You mentioned clearly in the Mortality section that Omicron related infections caused significantly higher mortality compared to non- Omicron, even though I do not see it in the Univariate and multivariate analysis table.!!

Thank you for your comment. We clarified the sentence that death was significantly higher in the non-Omicron group at the beginning of the Mortality section. We could not do univariate or multivariate analysis to compare them as the number of deaths was relatively small (33 total, only 6 in the Omicron group).

5. there was significant difference in all doses of vaccination between those who died and who did not, how did vaccination behave in the factor analysis.

Thank you for your comment. We added analysis for the first vaccination dose in both univariance and multivariate analysis in the results and mortality sections and Table 3. For the second and third vaccination doses, the number of vaccinated cases in the death group was very low to perform the analysis, 9 and 5 patients, respectively.

6. Please describe how COVID was managed in Qatar during the Omicron and on Omicron period, in ICU and general ward.

Thank you for your comment. We added a paragraph to the discussion section, including multiple references for further reading.

24- van Teijlingen E, Sathian B, Simkhada P, Banerjee I. COVID-19 in Qatar: Ways forward in public health and treatment. Qatar Med J. 2021 Jan 4;2020(3):38. doi: 10.5339/qmj.2020.38. PMID: 33447537; PMCID: PMC7780299.

25- Barman, M.; Hussain, T.; Abuswiril, H.; Illahi, M.N.; Sharif, M.; Saman, H.T.; Hassan, M.; Gaafar, M.; Abu, J.; Ahmad, M.K.K. Embracing Healthcare Delivery Challenges during a Pandemic. Review from a nodal designated COVID-19 center in Qatar. Avicenna 2021, 2021, 8.

26- Omrani, A.S., Almaslamani, M.A., Daghfal, J. et al. The first consecutive 5000 patients with Coronavirus Disease 2019 from Qatar; a nation-wide cohort study. BMC Infect Dis 20, 777 (2020). https://doi.org/10.1186/s12879-020-05511-8

27- Seedat, S., Chemaitelly, H., Ayoub, H.H., Makhoul, M., Mumtaz, G.R., Al Kanaani, Z., Al Khal, A., Al Kuwari, E., Butt, A.A., Coyle, P. and Jeremijenko, A., 2021. SARS-CoV-2 infection hospitalization, severity, criticality, and fatality rates in Qatar. Scientific reports, 11(1), pp.1-10.

28- Butt AA, Dargham SR, Tang P, Chemaitelly H, Hasan MR, Coyle PV, Kaleeckal AH, Latif AN, Loka S, Shaik RM, Zaqout A, Almaslamani MA, Al Khal A, Bertollini R, Abou-Samra AB, Abu-Raddad LJ. COVID-19 disease severity in persons infected with the Omicron variant compared with the Delta variant in Qatar. J Glob Health. 2022 Jul 6;12:05032. doi: 10.7189/jogh.12.05032. PMID: 35788085; PMCID: PMC9253930.

29- Khatib MY, Ananthegowda DC, Elshafei MS, et al.: Predictors of mortality and morbidity in critically ill COVID-19 patients: An experience from a low mortality country. Health Sci Rep. 2022, 5:e542. 10.1002/hsr2.542

30- AlNuaimi AA, Chemaitelly H, Semaan S, et al. All-cause and COVID-19 mortality in Qatar during the COVID-19 pandemic. BMJ Global Health 2023;8:e012291. doi:10.1136/ bmjgh-2023-012291

7.You mention use on monoclonal Ab and anti viral as possible explanation for the lower mortality in Omicron infected patients, was that standard regime of treatment of COVID during that time, how many patients received it?

Thank you for your comment. Antiviral treatment was included in the treatment protocol of all infected dialysis patients, given they were considered a high-risk group. On the other hand, monoclonal Ab treatment was initially used in the treatment protocol but was excluded later due to lack of efficacy. We do not have the accurate number of patients who received antiviral or monoclonal antibodies treatment as our study did not investigate the treatment details.

8. Your argument for the neutralizing Ab mounted from Omicron infection in the discussion is confusing, do you advocate that it contributed to lower mortality or not?

Thank you for your comment. We meant that the lower titers of Omicron-specific neutralizing antibodies compared to Delta-specific neutralizing antibodies might explain the rapid spread of Omicron infection but not the effect on mortality, as the virus might evade neutralizing antibodies induced by previous infection or vaccinations. We modified the text to clarify this point.

9. As Omicron caused much milder disease and lower mortality, on what basis do you advocate to deliver 4th booster dose.

Thank you for your comment. We mentioned that Cinkilik et al. (39) found an improved omicron-specific humoral immunity following the fourth mRNA vaccination. Although the Omicron variant resulted in mostly mild disease, the disease might still be severe in such a vulnerable dialysis population, and a fourth booster dose might be beneficial. We modified the language to clarify the statement in the revised manuscript.

10. The manuscript needs English language editing please. 

Extensive and professional language editing was done as advised.

---

## [Editor Report · Decision Letter 1]

10 Jul 2023

PONE-D-23-13125R1A comparative analysis of COVID-19 Omicron and non-Omicron variants in the hemodialysis Population of QatarPLOS ONE

Dear Dr. Hamad

Thank you for submitting your manuscript to PLOS ONE. After careful consideration, we feel that it has merit but does not fully meet PLOS ONE’s publication criteria as it currently stands. Therefore, we invite you to submit a revised version of the manuscript that addresses the points raised during the review process.

We look forward to receiving your revised manuscript.

Kind regards,

Fadi Aljamaan

Academic Editor

PLOS ONE

Additional Editor Comments:

Thanks for the revised version, but it did not improve much the quality of your manuscript and still there are major results/conclusions that needs addressing.

1. The manuscript still needs English language editing, for example “To compare characteristics of HD patients infected with COVID-19 who died to patients who stayed alive during the study and identify risk factors associated with death”

You can write it in better way : Assess characteristics of HD patients (Survivors Vs non survivors) who developed COVID-19 and identify factors associated with mortality.

2. “Per national genomic surveillance, all patients diagnosed with COVID-19 before December 1, 2021, were assigned to the non-Omicron group, while patients diagnosed on December 1, 2021, and onwards were assigned to the Omicron group”

Was the assignment to Omicron versus non omicron as per time frame or based on genomic testing of each case when diagnosed?!

3. How did you analyze the 26 patients who developed both Omicron and non Omicron variants?

4. How do you explain the same ICU length of stay between Omicron and Non Omicron infected patients even though the rate of mechanical ventilation was much higher in the Non Omicron group, why the Omicron infected patients were kept in the ICU?

5. What do you mean by univariate analysis in Table 3 and what is the difference between it and the multivariate analysis next to it and the difference with Table 2, and describe this in the methodology.

6. You mentioned the incidence of Omicron versus Non omicron variant infection in the discussion. How did you calculate it, and where is that information in the results section?

7. I don’t understand why do you mention stressfully the relation between CT value and outcome and correlate that with the Omicron values, even though the CT value was not statistically significant factor associated with mortality in the bivariate and multivariate analysis and the references you refer to do not give strong prediction of what you are proposing.

8. I still find your advocacy for the fourth dose of vaccine in HD patients specifically and in general is not strongly explained in the discussion part, as the available vaccines produce low level of Omicron-specific neutralizing antibodies, compared to other variants, and the Omicron variant is associated with very much mild disease, and facing the current worldwide discussion of possible side effects from COVID-19 vaccines that were fast tracked with now properly done studies, the fourth or subsequent vaccination for Omicron variant is not strongly justified.

9. Still you don’t explain the low mortality reasons in Omicron cases clearly in the discussion, once you claim it is the use of neutralizing Ab and then the antiviral, then the reception of booster vaccine dose (even though you claim that Omicron-specific neutralizing antibodies are not well mounted by vaccination), please improve the fidelity and strength of your discussion in that part.

---

## [Author Response · Author response to Decision Letter 1]

24 Jul 2023

Dear Editor in Chief

Thank you for your valuable comments. We did the recommended changes as advised. We hope this version will be satisfactory. Appreciate your efforts in improving our manuscript.

1. The manuscript still needs English language editing, for example “To compare characteristics of HD patients infected with COVID-19 who died to patients who stayed alive during the study and identify risk factors associated with death”

You can write it in better way : Assess characteristics of HD patients (Survivors Vs non survivors) who developed COVID-19 and identify factors associated with mortality.

Further English editing was reperformed as advised. We checked our manuscript against language editing service by AJE/Springer nature and it suggested that the paper do not need further editing (report attached with 92nd percentile). Please let us know if you feel further editing needed. 

2. “Per national genomic surveillance, all patients diagnosed with COVID-19 before December 1, 2021, were assigned to the non-Omicron group, while patients diagnosed on December 1, 2021, and onwards were assigned to the Omicron group”

Was the assignment to Omicron versus non omicron as per time frame or based on genomic testing of each case when diagnosed?!

Great comment. We added this short paragraph to the methods to explain with reference: 

Due to the fast rise of Omicron COVID-19 infections, large scale genome sequencing was not possible in the Omicron variant (compared to previous variants), and positive cases from December 2021 onwards were considered Omicron based on representative sampling.

Butt AA, Dargham SR, Tang P, Chemaitelly H, Hasan MR, Coyle PV, Kaleeckal AH, Latif AN, Loka S, Shaik RM, Zaqout A, Almaslamani MA, Al Khal A, Bertollini R, Abou-Samra AB, Abu-Raddad LJ. COVID-19 disease severity in persons infected with the Omicron variant compared with the Delta variant in Qatar. J Glob Health. 2022 Jul 6;12:05032. doi: 10.7189/jogh.12.05032. PMID: 35788085; PMCID: PMC9253930.

3. How did you analyze the 26 patients who developed both Omicron and non Omicron variants?

Cases were analyzed based on each episode of COVID as each episode carries its own different outcomes. We added this analysis in the methodology. 

4. How do you explain the same ICU length of stay between Omicron and Non Omicron infected patients even though the rate of mechanical ventilation was much higher in the Non Omicron group, why the Omicron infected patients were kept in the ICU?

Thank you for this observation. ICU admissions were due to many reasons. The percentage of patients who had mechanical ventilation out of all ICU admission were 43% in Omicron (3 out of 7) versus 63% in non-Omicron (28 out of 44). We had only a total of 7 patients in the Omicron group admitted to the ICU. Three of them were put on mechanical ventilation (table 1). The rest were put for other reasons (impending respiratory failure requiring Bipap, circulatory failure/hypotension/ sepsis). Will be happy to add more details in the discussion section if you deem necessary. Once there were in ICU, length of stay was similar as both groups had severe illness requiring admission.

5. What do you mean by univariate analysis in Table 3 and what is the difference between it and the multivariate analysis next to it and the difference with Table 2, and describe this in the methodology.

Thank you for this valuable comment. We added a paragraph in the methods to explain this in details.

6. You mentioned the incidence of Omicron versus Non omicron variant infection in the discussion. How did you calculate it, and where is that information in the results section?

Thank you for this observation. We agree with you. We added method of calculation in the methods section and outcome in the result section.

7. I don’t understand why do you mention stressfully the relation between CT value and outcome and correlate that with the Omicron values, even though the CT value was not statistically significant factor associated with mortality in the bivariate and multivariate analysis and the references you refer to do not give strong prediction of what you are proposing.

We agree with your comment. We rephrased the sentence to represent our significant outcomes only.

8. I still find your advocacy for the fourth dose of vaccine in HD patients specifically and in general is not strongly explained in the discussion part, as the available vaccines produce low level of Omicron-specific neutralizing antibodies, compared to other variants, and the Omicron variant is associated with very much mild disease, and facing the current worldwide discussion of possible side effects from COVID-19 vaccines that were fast tracked with now properly done studies, the fourth or subsequent vaccination for Omicron variant is not strongly justified.

Agree and removed this part from discussion.

9. Still you don’t explain the low mortality reasons in Omicron cases clearly in the discussion, once you claim it is the use of neutralizing Ab and then the antiviral, then the reception of booster vaccine dose (even though you claim that Omicron-specific neutralizing antibodies are not well mounted by vaccination), please improve the fidelity and strength of your discussion in that part.

Thank you for this important remark. We modified the discussion to highlight that transmission is higher in Omicron and potential causes (including evasion of vaccine), but we stressed that despite this evasion, there is still relative protection provided by the vaccine (especially with adding 3rd dose) compared to unvaccinated patients (and that include reducing transmission and morbidity). We added references too.

---

## [Editor Report · Decision Letter 2]

7 Aug 2023

PONE-D-23-13125R2A comparative analysis of COVID-19 Omicron and non-Omicron variants in the hemodialysis Population of QatarPLOS ONE

Dear Dr. Hamad,

Thank you for submitting your manuscript to PLOS ONE. After careful consideration, we feel that it has merit but does not fully meet PLOS ONE’s publication criteria as it currently stands. Therefore, we invite you to submit a revised version of the manuscript that addresses the points raised during the review process.

We look forward to receiving your revised manuscript.

Kind regards,

Fadi Aljamaan

Academic Editor

PLOS ONE

Journal Requirements:

Additional Editor Comments:

Dear authors

Thanks for your valuable response to the comments nd revised version you submitted.

I still have major concern about labelling the Omicron cases as you did not have definite objective method of confirming their diagnosis with the Omicron variant apart from the time variable (being diagnosed after December 2021) therefore I advise you to mention that as major limitation in your study and remove that from the title of the study as your definite results about the HD patients with COVID-19 mainly rather than Omicron versus non Omicron , and to transfer the Table of Omicron versus non Omicron to appendix due to the redundency of the inclusion criteria.

---

## [Author Response · Author response to Decision Letter 2]

24 Aug 2023

Dear Editor in Chief

Thank you for your valuable comment and feedback. Please find details reply to reviewers’ comments:

1- Please review your reference list to ensure that it is complete and correct. If you have cited papers that have been retracted, please include the rationale for doing so in the manuscript text, or remove these references and replace them with relevant current references. Any changes to the reference list should be mentioned in the rebuttal letter that accompanies your revised manuscript. If you need to cite a retracted article, indicate the article’s retracted status in the References list and also include a citation and full reference for the retraction notice.

All references were checked, and none were retracted. 

Style was checked and modified as needed.

2- I still have major concern about labelling the Omicron cases as you did not have definite objective method of confirming their diagnosis with the Omicron variant apart from the time variable (being diagnosed after December 2021) therefore I advise you to mention that as major limitation in your study and remove that from the title of the study as your definite results about the HD patients with COVID-19 mainly rather than Omicron versus non Omicron , and to transfer the Table of Omicron versus non Omicron to appendix due to the redundency of the inclusion criteria.

A- Your comment is very valuable and reasonable. We tried to follow lead of similar articles published in the literature (*). We agree with your assessment overall. We hope that we made the necessary changes to answer your query:

- We changed the title to reflect period of Omicron dominant period (instead of Omicron) to reflect on timing of COVID infection rather than viral gene sequencing as per your comment.

- We changes all Omicron to Omicron dominant period and We changed all non-Omicron to pre-Omicron period.

- We added it in major challenge as advised “First and most important, not all cases of Omicron dominant period are tested with genomic sequencing to prove the diagnosis. But sample testing shown that majority of COVID-19 cases in that period were of Omicron variant (24).”

- regarding table 1, as we clarified inclusion criteria (by periods not exact genomic testing). We modified title of columns to reflect that. We changed Omicron to Omicron dominant period and changed Non-Omicron to Pre-Omicron period. This table is important to reflect the changes in COVID-19 outcomes related to time (periods) and we feel it is important to the reader.

Overall, we made the changes per your advice and in similar way to describe time of infection with predominant variant rather than virus genome sequencing. 

Bohnert AS, Kumbier K, Rowneki M, Gupta A, Bajema K, Hynes DM, Viglianti E, O'Hare AM, Osborne T, Boyko EJ, Young-Xu Y, Iwashyna TJ, Maciejewski M, Schildhouse R, Dimcheff D, Ioannou GN. Adverse outcomes of SARS-CoV-2 infection with delta and omicron variants in vaccinated versus unvaccinated US veterans: retrospective cohort study. BMJ. 2023 May 23;381:e074521. doi: 10.1136/bmj-2022-074521. PMID: 37220941.

---

## [Editor Report · Decision Letter 3]

25 Aug 2023

From Past to Present: Exploring COVID-19 in Qatar's Hemodialysis Population across Omicron Dominant and pre-Omicron Periods

PONE-D-23-13125R3

Dear Dr. Abdullah Ibrahim Hamad

We’re pleased to inform you that your manuscript has been judged scientifically suitable for publication and will be formally accepted for publication once it meets all outstanding technical requirements.

Kind regards,

Fadi Aljamaan

Academic Editor

PLOS ONE
---

## [Editor Report · Acceptance letter]

6 Sep 2023

PONE-D-23-13125R3 

From Past to Present: Exploring COVID-19 in Qatar's Hemodialysis Population across Omicron Dominant and pre-Omicron Periods 

Dear Dr. Hamad:

I'm pleased to inform you that your manuscript has been deemed suitable for publication in PLOS ONE. Congratulations! Your manuscript is now with our production department. 

Kind regards, 

on behalf of

Dr. Fadi Aljamaan 

Academic Editor

PLOS ONE